# Pharmaceutical expenditure changes under the volume-based procurement policy: Effects and influencing factors

Ying Yang[1][⊙], Yuanhui Duan[1][⊙], Lei Zhou[1], Sisheng Gan[1], Zongfu Mao[2,3], Furong Wang[1,4]*

1 School of Nursing, Tongji Medical College, Huazhong University of Science and Technology, Wuhan, Hubei, China, 2 Global Health Institute, Wuhan University, Wuhan, Hubei, China, 3 Dong Fureng Institute of Economic and Social Development, Wuhan University, Wuhan, Hubei, China, 4 Department of Neurology, Tongji Hospital, Tongji Medical College, Huazhong University of Science and Technology, Wuhan, Hubei, China

⊙ These authors contributed equally to this work.
* wangfurong.china@163.com

## Abstract

### Objectives

To estimate the impact of China's volume-based procurement (VBP) policy on the expenditure of both policy-covered and uncovered drugs, and to identify the elements that contribute to drug expenditure changes under VBP policy.

### Methods

Using national drug procurement data of public medical institutions, this study included 25 policy-covered VBP drugs and 99 policy-uncovered alternative drugs as samples, seven "4+7" pilot cities and eight "4+7" expansion provinces as observation regions. Time-varying difference-in-difference (DID) model was applied to quantify policy impact on drug expenditures. The drug expenditure index decomposition method was employed to analyze the determinants of drug expenditure changes following VBP policy.

### Results

The expenditure of VBP drugs significantly decreased by 42.19% after VBP policy ($\beta = -0.55$, $p < 0.001$), while alternative drugs increased by 11.52% ($\beta = -0.11$, $p < 0.001$), with a significant reduction in the overall expenditure of observed drugs ($\beta = -0.05$, $p < 0.001$). The decrease of VBP drug expenditures showed a trend of tertiary hospital ($\beta = -0.64$, $p < 0.001$) > secondary hospital ($\beta = -0.57$, $p < 0.001$) > primary healthcare centers ($\beta = -0.39$, $p < 0.001$). The index decomposition showed that manufacturer structure index ($I_M$) decline was the primary driver for expenditure reduction of policy-covered drugs, with the $I_M$ decrease of 54.17% in pilot cities and

**Data availability statement:** The data that support the findings of this study are owned and managed by the National Health Commission (NHC) of the People's Republic of China and are not publicly available due to institutional policies. The authors do not have the authority to share the data. Data access may be granted upon reasonable request through the official website (https://www.nhc.gov.cn/) or contact number (+8610-68792114) of NHC, subject to institutional approval.

**Funding:** This study was supported by the National Natural Science Foundation of China (Grant number: 72404098), the China Postdoctoral Science Foundation (grant number: 2024M761028), the Postdoctor Project of Hubei Province (grant number: 2024HBBHCXB019), and the Postdoctoral Fellowship Program of CPSF (Grant number: GZC20240534).

**Competing interests:** The authors have declared that no competing interests exist.

**Abbreviations:** ATC, anatomical therapeutic chemical; CDSIP, China drug supply information platform; CNYChinese YuanDID, difference-in-difference; FE, fixed effects; GDP, gross domestic product; INN, international nonproprietary name; NHSA, National Healthcare Security Administration; PHCs, primary healthcare centers; VBP, volume-based procurement.

40.86% in expansion regions. The secondary driver was the price index ($I_P$), with a decline of 31.68% in pilot cities and 36.08% in expansion regions. The restraining factor was the quantity index ($I_Q$), increasing by 92.54% in pilot cities and 52.04% in expansion regions. $I_Q$ also drove the increase in alternative drug expenditures, increasing by 95.56% in pilot cities and 32.76% in expansion regions.

## Conclusion

VBP policy effectively promoted the decline of total drug expenditures, primarily through manufacturer-level market displacement and the absolute price reduction. However, the "spillover effect" of alternative drugs weakened the overall effect on cost control. Strengthening holistic governance and improving the quality and intensiveness of drug use are important directions for future policy perfection.

## Introduction

The inflated pricing of pharmaceuticals and rapid increase in drug expenditures have been widespread challenges faced by China and many low- and middle-income countries, leading to insufficient accessibility of high-quality medicines and a heavy economic burden on patients [1,2]. In China, the proportion of drug expenditures in total health expenditures was as high as 30%−40% during 2010–2020, which far exceeded the average level of OECD countries (14.9%) in 2020, as well as the United States (11.0%), Japan (18.1%), South Korea (20.1%) [3]. Since 2018, the Chinese government has launched the national volume-based procurement (VBP) policy adopting a "quantity-for-price" approach, to reduce drug prices, save drug expenditures, and achieve multi-dimensional reform goals of rational drug use in medical institutions and high-quality development of the pharmaceutical industry [4–6].

Literature has demonstrated the effect of the VBP policy on reducing policy-covered drug expenditures. Firstly, utilizing the macro-level drug procurement data, previous studies reported a significant decrease in the procurement expenditure of VBP drugs after policy implementation [7–12]. Secondly, employing the micro-level medical data, relevant studies indicated the decline of per-visit and per-admission drug costs after the VBP policy among patients treated with bid-winning drugs, or specific diseases and medical institutions where VBP drugs were centralized consumed [13–17]. Due to the common "clinical substitutability" characteristics of drug use, researchers have also paid attention to the expenditure change of drugs that are outside the VBP list but have a substitutive relationship with VBP policy-covered drugs, yet relevant findings remain contentious. For example, Chen et al. [18] reported increases in the procurement expenditure of basic alternative drugs after the VBP policy through the expenditure index decomposition analysis, highlighting the existence of a "spillover effect". However, Ma [19] employed the difference-in-difference (DID) model and found that the VBP policy implementation significantly reduced procurement expenditure of alternative drugs by 18.7%, in line with the substitution effect of VBP drugs on their completely alternative drugs

proposed by Liu et al. [20]. Meanwhile, several studies reported no significant impact of VBP policy on the expenditure of alternative INNs [9,10,12].

The factors influencing drug expenditures are complex and diverse [21], which are related to the overall effectiveness and sustainability of the pharmaceutical reform. Zhou et al. [8] examined the influence factors of the cost-control effect of VBP policy from the perspective of regional policy implementation employing the qualitative comparative analysis method, highlighting the roles of regional medical insurance fund balance, fiscal health expenditure, and policy organization experience. The more direct driving force behind the change in drug expenditures is the prescribing behavior and structure [22], so it is the focus of the intervention for cost-containment policies. A study on the pharmaceutical cost-control effect of Sanming medical reform revealed that the trend of hospitals' preference for high-priced drugs was effectively curbed after policy implementation, thereby achieving effective control over drug expenditures [23].

In summary, the large number of existing research on drug expenditures under VBP policy implementation, on the one hand, lack reliable evidence when involving the scope of drugs not covered by the policy; on the other hand, no studies have systematically investigated the elements influencing drug expenditure changes from the perspective of hospital drug use. This study tries to make up for the above two aspects. Therefore, we employed the DID model and the drug expenditure index decomposition method, firstly, to estimate the impact of VBP policy on the expenditure of both VBP policy-covered drugs and uncovered alternative drugs, and secondly, to identify the key elements that contribute to drug expenditure changes under the VBP policy. The findings aspire to provide insights and references for the refinement of China's VBP policy and the pharmaceutical practice in other developing countries.

## Materials and methods

### Data sources

The data utilized in this study were derived from the China Drug Supply Information Platform (CDSIP), which provides access to drug procurement and use order data from the provincial drug centralized bidding and procurement platform of 31 provinces (autonomous regions and municipalities) in mainland China. The data extracted from the CDSIP database include the name of the medical institution, procurement date, drug product code, drug generic name, dosage form, specification, package, manufacturing enterprise, price per unit, purchasing unit (by box, bottle, or branch), purchase volume, purchase expenditures, etc.

This study extracted data spanning a total of 36 months from January 2018 to December 2020, with the month serving as the smallest observational time unit, encompassing 36 observation time points. This study considered the implementation of the first VBP batch consisting of two policy rounds – "4+7" pilot (implemented in 11 cities) and "4+7" expansion (implemented in 25 regions). To ensure data integrity, seven "4+7" pilot cities and eight "4+7" expansion provinces were included as the observation region in this study. The basic information of the regions is detailed in S1 Table.

### Samples

This study included the policy-covered drugs in the first VBP batch and their alternative drugs that have not yet been covered by VBP policy as the overall observation samples.

(1) VBP drugs. It represents the scope of drugs covered by the VBP policy. We included drugs by international nonproprietary name (INN) that are listed in the VBP list and have been successfully procured. Information related to VBP INN is publicly available on the website of the Joint Procurement Office (https://www.smpaa.cn). A total of 25 VBP INNs in the first policy batch were included, covering 55 drug generic names, 1025 drug products, and involving 228 drug manufacturers.

(2) Alternative drugs. It represents the scope of drugs related to VBP policy but not directly covered by the policy. We included the drug INN that shares the same therapeutic category as the VBP INNs and has a clinical substitutability

relationship with them. In this study, the inclusion of alternative INNs was determined following the "list of alternative drugs" provided by the National Healthcare Security Administration (NHSA) in the *Monitoring Plan for National Centralized Volume-based Drug Procurement and Use Pilot Work*. The list of alternative drugs is formulated by the NHSA based on clinical substitution relationships reported by expert clinicians, while also taking into account actual procurement and use substitution patterns reflected in dynamic procurement data [24]. Among them, five drugs (Simvastatin, Olmesartan, Cefalexin, Adefovir, and Candesartan) that were included in the second VBP batch during the observation period were excluded from the observation sample, to eliminate the impact of other VBP batches' implementation. A total of 99 alternative INNs were included in the analysis, covering 256 drug generic names, 3523 drug products, and involving 676 drug manufacturers.

VBP drugs and alternative drugs respectively represent the scope of drugs covered and uncovered by the VBP policy, which is helpful for observing the direct and indirect impacts of policy implementation. Ultimately, 311 drug generic names and 4548 drug products were included in the analysis, involving 760 drug manufacturers. According to the Anatomical Therapeutic Chemical (ATC), the observed drugs covered 8 anatomical categories and 12 therapeutic subcategories. Detailed information on the included drugs is provided in S2 Table.

## Data analysis

This study adopts a comparative analysis approach overall. Firstly, in the time dimension, we divided time period before and after policy intervention to quantify changes and impacts caused by VBP policy implementation. Secondly, in the region dimension, we distinguished between "4+7" pilot cities and expansion regions, which represent two policy implementation phases with different implementation timelines, necessitating separate assessments and comparisons. Thirdly, in the drug scope dimension, we included VBP drugs and their alternative drugs, to simultaneously observe and compare the direct and indirect effects of the policy. The following methods are mainly employed.

## Difference-in-difference model

This study employed a difference-in-differences model to estimate the effect of the VBP policy on drug expenditures. The drug expenditure indicator specifically refers to medical institutions' procurement expenditures of a defined range of drugs, which is distinct from the medication costs of individual patients occurred during outpatient or inpatient visits. However, under the zero-markup policy that mandates identical procurement and retail prices for drugs in public medical institutions [25], and given procurement regulations requiring all drugs to be procured through provincial drug centralized bidding and procurement platform [26], drug procurement expenditures can essentially be considered as the aggregate sum of medication costs across all patient visits. Following the fundamental principle of expenditure equals price multiplied by quantity, the expenditure ($E_{it}$) of drug $i$ during the observation period $t$ was calculated as the product of the price ($P_{it}$) and quantity ($Q_{it}$). The formula is expressed as follows:

$$E_{it} = \sum_{t=1}^{n} \sum_{i=1}^{n} P_{it} \times Q_{it}$$

(1)

Given that the first VBP batch was conducted in two rounds, with the first round ("4+7" pilot) implemented in pilot cities in March 2019, and the second round ("4+7" expansion) rolled out in other regions excluding the pilot cities (referred to as the expansion regions) in December 2019. The procurement drug list was consistent across the two rounds, yet the timing of policy implementation varied between regions. Under such a situation, the standard DID model is inapplicable due to the inability to separate a "clean" treatment group and treatment period [27]. Therefore, this study adopted a time-varying DID approach, utilizing the implementation of the first VBP batch of VBP policy as the treatment, considering both the

pilot cities and the expansion regions as the common observation area, constructing an interaction term ($D_{it}$) between treatment group and treatment period, which took the value of 1 if a region was subject to VBP policy intervention during a certain period, and 0 otherwise. Specifically, in pilot cities, $D_{it}$ was coded 0 from January 2018 to February 2019 and 1 from March 2019 to December 2020; and in expansion regions, it was coded 0 from January 2018 to November 2019 and 1 from December 2019 to December 2020. The empirical model is as follows:

$$Y_{ijt} = \alpha + \beta D_{it} + \gamma X_i + \delta sin(2\pi T/12) + \sigma cos(2\pi T/12) + \mu_i + \theta_j + \varepsilon_{jit} \tag{2}$$

Where, $Y_{ijt}$ is the explained variable referring to drug expenditures, and is incorporated into the model after a logarithmic transformation. $D_{it}$ is a binary dummy variable indicating the policy intervention, with its coefficient $\beta$ representing the effect of VBP policy on drug expenditures. $X_i$ constituted a set of control variables accounting for differences in economic, population, and medical services [28], including per capita regional gross domestic product (GDP), the number of population at the year-end, and the number of clinical visits to medical institutions, which are also log-transformed before included in the model. Following previous research practice [29,30], this study controlled seasonal fluctuations of drug expenditures using the sine and cosine functions of time, denoted as $sin(2\pi T/12)$ and $cos(2\pi T/12)$, respectively. $\mu_i$, $\theta_j$, and $\delta_t$ denoted the fixed effects (FE) for the observation region (region FE), the drug (INN FE), and the time (month FE), respectively. $\varepsilon_{jit}$ is the random error term.

Subsequently, subgroup analyses were conducted based on the types of medical institutions and the drug ATC categories to assess the heterogeneity of analysis results across different subgroups. We categorized public medical institutions into tertiary hospitals, secondary hospitals, and primary healthcare centers (PHCs) in accordance with China's hierarchical classification regulations for medical institutions, as differences typically exist among these institution types in terms of drug formulary, medication scale, and prescribing habits. We stratified observation drugs into distinct therapeutic areas based on the first-level ATC codes, given that drug demand and market dynamics usually vary across therapeutic areas. The robustness test was conducted by three approaches: first, parallel trend test; second, standard DID analysis with pilot cities serving as the treatment group and expanded regions as the control group; and third, time-varying DID analysis by excluding potential interference time points.

## Index decomposition method on drug expenditure

The index decomposition method is conducive to discerning the underlying reasons for changes in drug expenditure changes following policy intervention. The most commonly used is the three-factor decomposition method, also known as the A.M index [31], which decomposes the overall change in drug expenditure ($I_E$) into three components: the price index ($I_P$), the quantity index ($I_Q$), and the structure index ($I_S$), as follows:

$$I_E = I_P \times I_Q \times I_S = \frac{\sum P_1 Q_1}{\sum P_0 Q_0} = \frac{\sum P_1 Q_0}{\sum P_0 Q_0} \times \frac{\sum Q_1}{\sum Q_0} \times \frac{\frac{\sum P_1 Q_1}{\sum Q_1}}{\frac{\sum P_1 Q_0}{\sum Q_0}} \tag{3}$$

Based on the three-factor decomposition method, Han et al. [32] further decomposed the structure index into three distinct components: chemical class structure ($I_C$), drug structure ($I_D$), and manufacturer structure ($I_M$) (Fig 1). The subscript $C$ refers to the chemical class determined based on the ATC code, where drugs sharing the same first five digits of the ATC code are classified into the same chemical class. The subscript $D$ refers to the drug's generic name. The subscript $M$ refers to the manufacturing enterprise. The calculation method is as follows:

$$I_E = \frac{\sum P_1 Q_1}{\sum P_0 Q_0} = \frac{\sum P_{M1} Q_{M0}}{\sum P_{M0} Q_{MO}} \times \frac{\sum Q_{M1}}{\sum Q_{M0}} \times \frac{\frac{\sum P_{C1} Q_{C1}}{\sum Q_{C1}}}{\frac{\sum P_{C1} Q_{C0}}{\sum Q_{C0}}} \times \left( \frac{\frac{\sum P_{D1} Q_{D1}}{\sum Q_{D1}}}{\frac{\sum P_{D1} Q_{D0}}{\sum Q_{D0}}} \middle/ \frac{\frac{\sum P_{C1} Q_{C1}}{\sum Q_{C1}}}{\frac{\sum P_{C1} Q_{C0}}{\sum Q_{C0}}} \right) \times$$

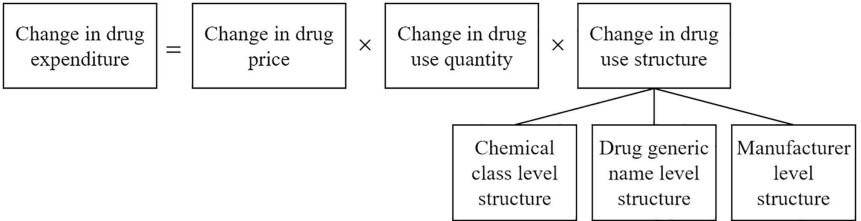

**Fig 1. Schematic diagram for the decomposition of drug expenditure change determinants.**

$$\left( \frac{\frac{\sum P_{M1}Q_{M1}}{\sum Q_{M1}}}{\frac{\sum P_{M1}Q_{M0}}{\sum Q_{M0}}} \Big/ \frac{\frac{\sum P_{D1}Q_{D1}}{\sum Q_{D1}}}{\frac{\sum P_{D1}Q_{D0}}{\sum Q_{D0}}} \right)$$

(4)

In this study, we employed Han et al.'s [32] modified drug expenditure index decomposition method. The period between January and June 2018 was assigned as the baseline period ($T_0$) for index decomposition, and the remaining observation period is segmented into three stages according to the implementation time of the "4+7" pilot and "4+7" expansion: $T_1$ (July 2018 to February 2019), $T_2$ (March to November 2019), and $T_3$ (December 2019 to December 2020), respectively. Among them, $T_0$ and $T_1$ represent the pre-intervention periods of "4+7" pilot, while $T_2$ and $T_3$ denote the post-intervention periods; $T_0$, $T_1$, and $T_2$ represent the pre-intervention periods of "4+7" expansion, with $T_3$ marking the post-intervention period. We designed multiple observation periods prior to policy implementation to monitor pre-intervention trends and mitigate potential confounding factors beyond the VBP policy.

## Results

### Descriptive analysis

Table 1 describes the expenditure and expenditure structure of the observed drugs. After the VBP policy, the total expenditure of observed drugs in pilot cities and expansion regions decreased by 16.32% and 40.80%, among which the expenditure of VBP INNs decreased by 47.46% and 60.41%. When stratified by the type of medical institutions – tertiary hospitals, secondary hospitals, and PHCs, the total expenditure of observed drugs in pilot cities declined by 13.28%, 16.09%, and 21.74%, while in expansion regions, the decrease was 43.33%, 38.26%, and 35.55%. The reduction of VBP INN expenditures for tertiary hospitals, secondary hospitals, and PHCs was 40.13%, 49.98%, and 60.52% in the pilot cities, and 61.38%, 60.21%, and 56.29% in the expansion regions. As for the alternative INNs, the expenditure increased by 24.85% in the pilot cities but decreased in the expansion regions (−14.88%). Through in-depth region-specific observation (S3 **and** S4 Tables, S1 Fig), the decline in alternative INN expenditures only occurred in a few expansion provinces (Jiangsu, Hubei, and Hunan).

Fig 2 illustrates the monthly trend of drug expenditure in the observation regions. The trend chart visually showed a significant drop in the expenditure of VBP INNs around March 2019 for the pilot cities and November 2019 for the expansion regions, with a corresponding notable decrease in total expenditure of both VBP and alternative INNs during these time points. In the post-VBP policy period, the drug expenditure generally remained stable.

### The results of DID estimation

Table 2 presents the DID modeling results predicting the impact of VBP policy on the expenditure of VBP INNs, alternative INNs, and total observed drugs. The total expenditure of observed drugs decreased significantly ($\beta=-0.05$, $p<0.001$), with a prominent reduction of 42.19% in the expenditure of VBP INNs ($\beta=-0.55$, $p<0.001$), while the expenditure of alternative INNs significantly increased by 11.52% ($\beta=0.11$, $p<0.001$).

**Table 1. Descriptive changes of drug expenditure under VBP policy (billion CNY).**

| Category | Pilot cities[a] | | | Expansion regions[b] | | |
|---|---|---|---|---|---|---|
| | Pre-VBP (billion CNY) | Post-VBP (billion CNY) | GR (%) | Pre-VBP (billion CNY) | Post-VBP (billion CNY) | GR (%) |
| **Overall** | | | | | | |
| VBP INNs | 6.63 | 3.48 | −47.46 | 9.67 | 3.83 | −60.41 |
| Alternative INNs | 5.01 | 6.26 | 24.85 | 7.31 | 6.23 | −14.88 |
| Total | 11.64 | 9.74 | −16.32 | 16.98 | 10.05 | −40.80 |
| **Tertiary hospitals** | | | | | | |
| VBP INNs | 3.62 | 2.17 | −40.13 | 6.25 | 2.41 | −61.38 |
| Alternative INNs | 2.37 | 3.03 | 27.71 | 4.03 | 3.41 | −15.39 |
| Total | 5.99 | 5.20 | −13.28 | 10.28 | 5.83 | −43.33 |
| **Secondary hospitals** | | | | | | |
| VBP INNs | 1.21 | 0.60 | −49.98 | 2.06 | 0.82 | −60.21 |
| Alternative INNs | 0.98 | 1.23 | 25.48 | 1.32 | 1.27 | −3.94 |
| Total | 2.19 | 1.84 | −16.09 | 3.38 | 2.09 | −38.26 |
| **PHCs** | | | | | | |
| VBP INNs | 1.80 | 0.71 | −60.52 | 1.36 | 0.59 | −56.29 |
| Alternative INNs | 1.66 | 1.99 | 20.40 | 1.96 | 1.54 | −21.19 |
| Total | 3.46 | 2.71 | −21.74 | 3.32 | 2.14 | −35.55 |

*Note:* [a]The pre- and post-VBP periods for "4+7" pilot cities are March to December 2018 and March to December 2019. [b]The pre- and post-VBP periods for "4+7" expansion regions are January to November 2019 and January to November 2020.

VBP, volume-based procurement; CNY, Chinese yuan; GR, growth rate; INN, international nonproprietary name; PHCs, primary healthcare centers.

Subgroup analysis by the type of medical institutions revealed that policy impacts on the expenditure of VBP INNs across different medical institution types were uniformly and significantly negative (all $p$-values<0.05), with reduction of 47.22%, 43.28%, and 32.29% for tertiary hospitals, secondary hospitals, and PHCs, respectively. Conversely, policy impacts on the expenditure of alternative INNs across different medical institution types were significantly positive (all $p$-values<0.05), with increments of 6.82%, 11.96%, and 18.89% for tertiary hospitals, secondary hospitals, and PHCs, respectively. As for the total expenditure of VBP INNs and alternative INNs, a significant decrease was observed in tertiary and secondary hospitals (all $p$-values<0.001), while no significant change was detected in the PHCs ($p$>0.05). Subgroup analysis by the ATC categories showed that, except for the category L, the expenditure of VBP INNs in all categories significantly decreased (all $p$-values<0.05); the categories C, N, and L exhibited a significant increase in the expenditure of alternative INNs after policy intervention (all $p$-values<0.05).

The robustness of the DID estimation results was tested in the following three approaches. First, the results of the parallel trend test indicated that, in the pre-VBP period, the pilot cities and the expansion regions were comparable in terms of changes in drug expenditures (log-transformed), meeting the prerequisite condition for applying the DID method (S5-S7 Tables). Second, we conducted a standard DID analysis with the pilot cities as the treatment group and the expansion regions as the control group using the data from January 2018 to November 2019, and the results were generally consistent with the time-varying DID estimation (S8 Table). Third, we conducted time-varying DID estimation by excluding two potentially confounding time points (March and April 2019), the results were generally robust (S9 Table).

## The results of expenditure index decomposition

The results of drug expenditure index decomposition stratified by observation region, observation period, and the type of drug are shown in **Fig 3** and **Table 3**.

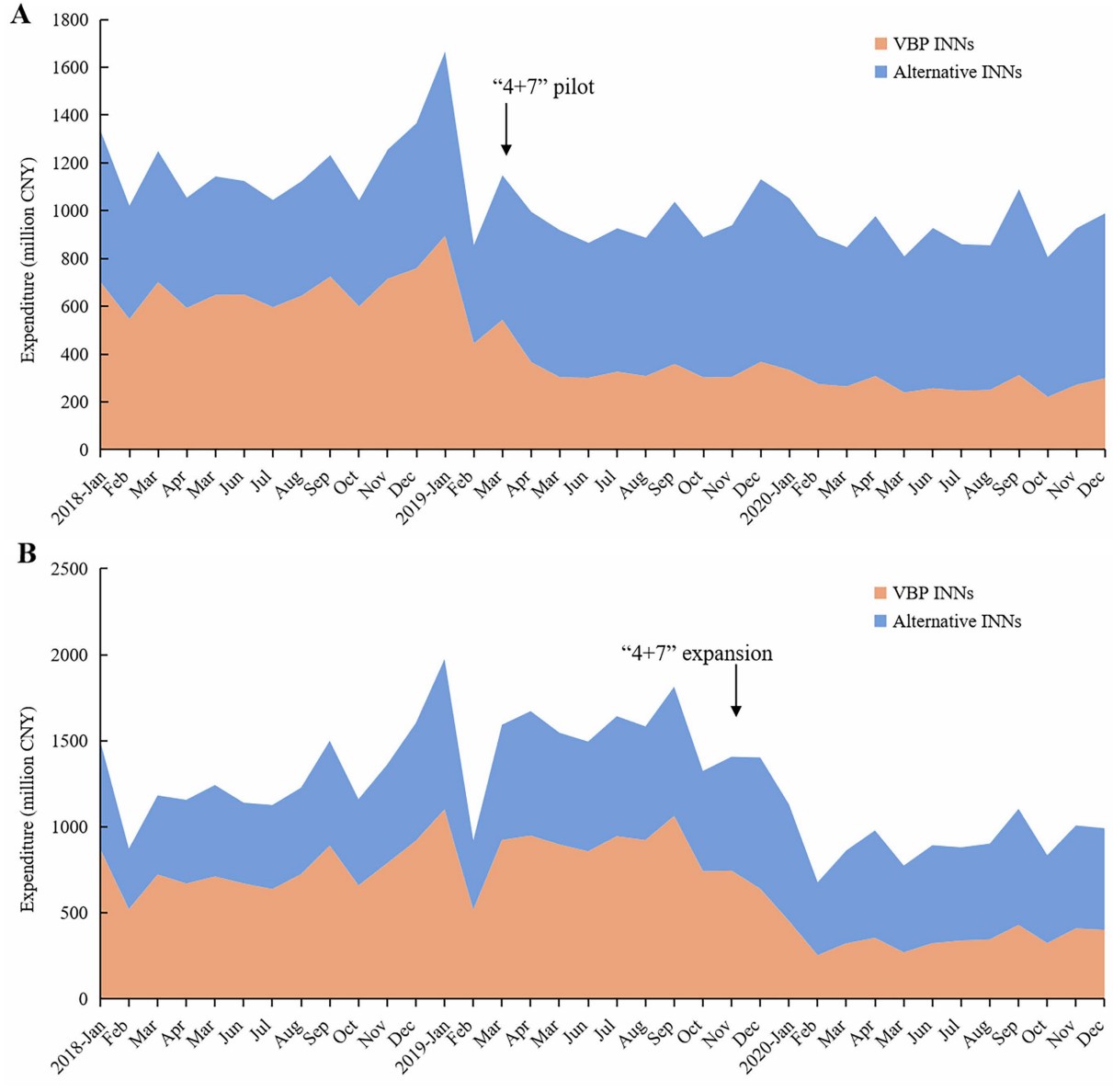

**Fig 2. Monthly trend of drug expenditures for VBP INNs and alternative INNs under VBP policy.** (A) Pilot cities, (B) Expansion regions. *Note*: VBP, volume-based procurement; INN, international nonproprietary name.

Firstly, prior to the implementation of the VBP policy, the expenditure index of both VBP INNs and alternative INNs in the observation regions exhibited an upward trend. The expenditure index of VBP INNs and alternative INNs, in the pilot cities, increased from 1.00 to 1.40 and 1.36, respectively; and in the expansion regions, rose from 1.00 to 1.91 and 2.05. During these periods, the primary driver for the increase in the expenditure index was the rise in the quantity index, which increased from 1.00 to 1.34 and 1.35 in pilot cities, and from 1.00 to 1.96 and 1.74 in the expansion regions.

Secondly, after policy implementation, the expenditure index of VBP INNs dropped from 1.40 to 0.86 (−38.57%) in pilot cities, and from 1.91 to 1.08 (−43.46%) in expansion regions. The primary contributor was the decline of the structure index, which decreased from 1.03 to 0.48 (−53.40%) in the pilot cities and from 1.00 to 0.58 (−42.00%) in the expansion

**Table 2. DID results for the impact of VBP policy on drug expenditures.**

| Category | VBP INNs | | | Alternative INNs | | | All observed drugs | | |
|---|---|---|---|---|---|---|---|---|---|
| | Coefficient | P-value | 95% CI | Coefficient | P-value | 95% CI | Coefficient | P-value | 95% CI |
| **Total** | −0.55 | 0.000 | −0.59 to −0.51 | 0.11 | 0.000 | 0.07 to 0.14 | −0.05 | 0.000 | −0.08 to-0.03 |
| **Medical institution type** | | | | | | | | | |
| Tertiary hospital | −0.64 | 0.000 | −0.68 to −0.60 | 0.07 | 0.000 | 0.03 to 0.10 | −0.11 | 0.000 | −0.14 to −0.09 |
| Secondary hospital | −0.57 | 0.000 | −0.63 to −0.51 | 0.11 | 0.000 | 0.07 to 0.16 | −0.08 | 0.000 | −0.12 to −0.05 |
| PHCs | −0.39 | 0.000 | −0.47 to −0.31 | 0.17 | 0.000 | 0.11 to 0.24 | 0.01 | 0.826 | −0.05 to 0.06 |
| **Therapeutic category** | | | | | | | | | |
| ATC_C | −0.72 | 0.000 | −0.79 to −0.65 | 0.26 | 0.000 | 0.19 to 0.32 | 0.01 | 0.855 | −0.05 to 0.06 |
| ATC_N | −0.25 | 0.000 | −0.33 to −0.17 | 0.11 | 0.000 | 0.07 to 0.16 | 0.05 | 0.020 | 0.01 to 0.09 |
| ATC_L | −0.02 | 0.742 | −0.14 to 0.10 | 0.20 | 0.002 | 0.07 to 0.33 | 0.10 | 0.044 | 0.003 to 0.19 |
| ATC_J | −1.03 | 0.000 | −1.12 to −0.94 | −0.26 | 0.000 | −0.36 to −0.17 | −0.48 | 0.000 | −0.56 to −0.41 |
| Others | −0.64 | 0.000 | −0.74 to −0.55 | 0.01 | 0.759 | −0.08 to 0.10 | −0.18 | 0.000 | −0.25 to −0.10 |

*Note:* VBP, volume-based procurement; INN, international nonproprietary name; CI, confidence interval; PHCs, primary healthcare centers; ATC, anatomical therapeutic and chemical.

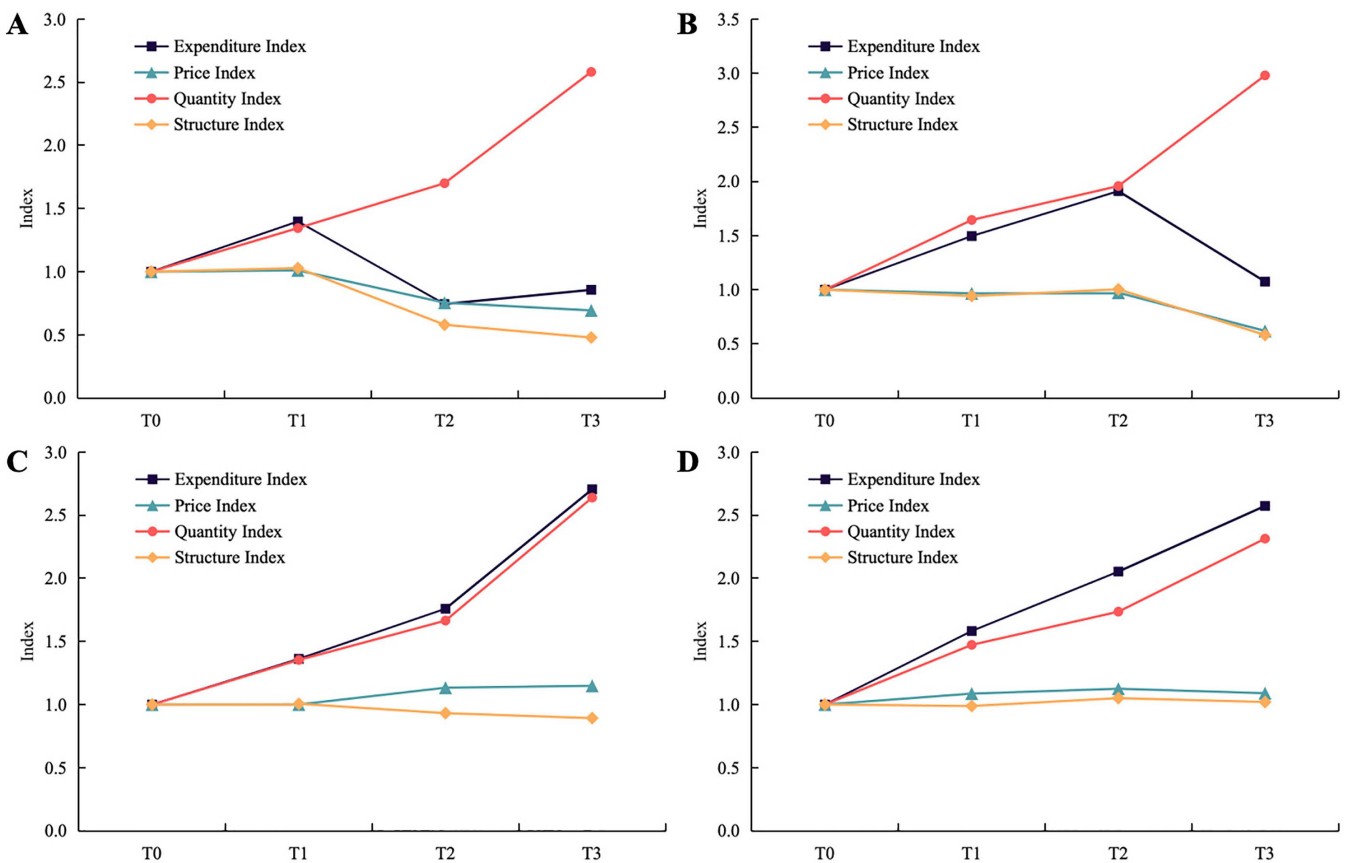

**Fig 3. Drug expenditure decomposition analysis for (A) VBP INNs in pilot cities, (B) VBP INNs in expansion regions, (C) alternative INNs in pilot cities, and (D) alternative INNs in expansion regions.** *Note:* $T_0$: January to June 2018; $T_1$: July 2018 to February 2019; $T_2$: March to November 2019; $T_3$, December 2019 to December 2020. For the pilot cities, $T_2$ and $T_3$ are the post-VBP periods; for the expansion regions, $T_3$ is the post-VBP period.

**Table 3. Exponential decomposition results regarding influence factors of drug expenditure change under VBP policy.**

| Category | Pilot cities | | | | Expansion regions | | | |
|---|---|---|---|---|---|---|---|---|
| | $T_0$ | $T_1$ | $T_2$ | $T_3$ | $T_0$ | $T_1$ | $T_2$ | $T_3$ |
| **VBP INNs** | | | | | | | | |
| Expenditure index ($I_E$) | 1.00 | 1.40 | 0.74 | 0.86 | 1.00 | 1.50 | 1.91 | 1.08 |
| Price index ($I_P$) | 1.00 | 1.01 | 0.75 | 0.69 | 1.00 | 0.97 | 0.97 | 0.62 |
| Quantity index ($I_Q$) | 1.00 | 1.34 | 1.70 | 2.58 | 1.00 | 1.64 | 1.96 | 2.98 |
| Structure index ($I_S$) | 1.00 | 1.03 | 0.58 | 0.48 | 1.00 | 0.94 | 1.00 | 0.58 |
| Chemical class structure ($I_C$) | 1.00 | 1.06 | 1.16 | 1.09 | 1.00 | 0.99 | 1.07 | 1.05 |
| Drug structure ($I_D$) | 1.00 | 1.00 | 1.01 | 1.01 | 1.00 | 1.00 | 1.00 | 1.01 |
| Manufacturer structure ($I_M$) | 1.00 | 0.96 | 0.50 | 0.44 | 1.00 | 0.95 | 0.93 | 0.55 |
| **Alternative INNs** | | | | | | | | |
| Expenditure index ($I_E$) | 1.00 | 1.36 | 1.76 | 2.71 | 1.00 | 1.58 | 2.05 | 2.57 |
| Price index ($I_P$) | 1.00 | 1.00 | 1.13 | 1.15 | 1.00 | 1.09 | 1.13 | 1.09 |
| Quantity index ($I_Q$) | 1.00 | 1.35 | 1.67 | 2.64 | 1.00 | 1.47 | 1.74 | 2.31 |
| Structure index ($I_S$) | 1.00 | 1.01 | 0.93 | 0.89 | 1.00 | 0.99 | 1.05 | 1.02 |
| Chemical class structure ($I_C$) | 1.00 | 1.00 | 1.04 | 1.03 | 1.00 | 1.05 | 1.10 | 1.06 |
| Drug structure ($I_D$) | 1.00 | 1.02 | 1.04 | 1.03 | 1.00 | 1.02 | 1.03 | 1.02 |
| Manufacturer structure ($I_M$) | 1.00 | 0.99 | 0.87 | 0.84 | 1.00 | 0.91 | 0.93 | 0.94 |
| **All observed drugs** | | | | | | | | |
| Expenditure index ($I_E$) | 1.00 | 1.38 | 1.20 | 1.69 | 1.00 | 1.64 | 2.14 | 2.23 |
| Price index ($I_P$) | 1.00 | 1.01 | 0.91 | 0.89 | 1.00 | 1.09 | 1.13 | 1.09 |
| Quantity index ($I_Q$) | 1.00 | 1.35 | 1.68 | 2.62 | 1.00 | 1.54 | 1.82 | 2.57 |
| Structure index ($I_S$) | 1.00 | 1.02 | 0.78 | 0.73 | 1.00 | 0.98 | 1.04 | 0.80 |
| Chemical class structure ($I_C$) | 1.00 | 1.04 | 1.05 | 1.03 | 1.00 | 1.03 | 1.10 | 1.05 |
| Drug structure ($I_D$) | 1.00 | 1.00 | 1.05 | 1.05 | 1.00 | 1.02 | 1.02 | 1.00 |
| Manufacturer structure ($I_M$) | 1.00 | 0.98 | 0.71 | 0.68 | 1.00 | 0.93 | 0.93 | 0.76 |

VBP, volume-based procurement; INN, international nonproprietary name.

*Note:* $T_0$: January to June 2018; $T_1$: July 2018 to February 2019; $T_2$: March to November 2019; $T_3$: December 2019 to December 2020. For the pilot cities, $T_2$ and $T_3$ are the post-VBP periods; for the expansion regions, $T_3$ is the post-VBP period.

regions. As well as the change in the structure index was mainly due to the manufacturer-level structure. The secondary contributor was the decrease in the price index, which dropped from 1.01 to 0.69 (−31.68%) in pilot cities and from 0.97 to 0.62 (−36.08%) in expansion regions. The restraining factor was the increase in quantity index, which increased from 1.34 to 2.58 (92.54%) in pilot cities and from 1.96 to 2.98 (52.04%) in expansion regions.

Thirdly, in the post-VBP periods, the expenditure index of alternative INNs increased from 1.36 to 2.71 (99.26%) in pilot cities and from 2.05 to 2.57 (25.37%) in expansion regions. The main driver was the increase in the quantity index, which increased from 1.35 to 2.64 (95.56%) in pilot cities and from 1.74 to 2.31 (32.76%) in expansion regions.

## Discussion

### VBP policy promoted the reduction of drug expenditures

This study initially found that the VBP policy significantly reduced the expenditure of policy-covered VBP INNs, which is consistent with previous study findings [8,13,33]. The overall procurement expenditure of observed drugs within the same therapeutic category as VBP INNs also significantly decreased under the impact of the VBP policy. Although this

study used macro-level aggregated data rather than micro-level patient data, under the zero-markup policy, the significant reduction in total procurement expenditure can point to savings in patient medication costs. For example, several studies utilizing individual patient data have reached similar conclusions [14–16,34,35], indicating that VBP policy can indeed bring benefits in reducing the medication burden of patients.

We also observed a significant increase in the expenditure of alternative INNs following the VBP policy, suggesting an inter-drug "spillover effect", which is consistent with the findings of Chen et al. [18] and Yang [36]. It was estimated that [36], approximately one-third of the drug expenditure savings expected from the VBP INNs were offset by the increase in the expenditure of alternative INNs, weakening the overall cost-saving effect of the VBP policy.

Additionally, subgroup analysis found that the impact of VBP policy on drug expenditure varied by region, medical institution type, and drug category. Firstly, the reduction of drug expenditure in expansion regions is slightly higher than that in pilot cities, which might be related to the relatively lagging development of the expansion regions, where residents have a more urgent need for low-priced drugs [37]. Secondly, the reduction of drug expenditure in tertiary hospitals and secondary hospitals was more prominent than PHCs, consistent with Chen et al.'s [7] comparative study on PHCs and secondary or above hospitals. This may be related to the larger hospital scale and greater drug expenditure base in tertiary and secondary hospitals, thus offering greater room for expenditure savings, and may also be associated with a far greater drug use increment in primary healthcare settings under the VBP policy [20,37,38]. Thirdly, the change in drug expenditures varied across different therapeutic categories. For example, unlike other categories, there was no significant decrease in the expenditure of VBP anti-cancer drugs after policy implementation, which may be related to the rapid growth of anti-cancer drug consumption following the VBP policy [39,40].

## Manufacturer structure is the primary driver for drug expenditure reduction under the VBP policy

Previous literature has indicated that structural factors are the main driver of rapid drug expenditure increment [22,23,41], reflecting the preference of physicians and patients for more expensive drugs. In this study, the results of index decomposition revealed that the primary driver for the reduction of policy-covered drug expenditures following the VBP policy was the sharp decline of the drug structure index, not only in the first round "4+7" pilot cities (dropped from 1.03 to 0.48) but also in the second round "4+7" expansion regions (dropped from 1.00 to 0.58). The results are generally consistent with a previous observation in Shenzhen, China [18]. Furthermore, among the three structural dimensions of chemical category, drug generic name, and manufacturing enterprise, only the manufacturer structure was found to have an absolute contribution to the decrease in the expenditure of VBP INNs.

In the national-level VBP policy, one of the key policy measures is to take generics consistency evaluation (GCE) on quality and efficacy as a criterion for drug quality access, intensifying market competition between GCE-certificated generic drugs and original brand-name drugs to reduce drug prices. The reason why VBP policy promotes expenditure savings by optimizing manufacturer structure is twofold: on the one hand, GCE-uncertificated generic drugs are not eligible to participate in VBP bidding, leading to the gradual withdrawal of certain products and corresponding manufacturers from the "mainstream" hospital-end drug market [42,43]; on the other hand, the VBP policy inevitably lead to large market displacement of bid-winning products over bid-non-winning products, most notably the substantial replacement upon bid-non-winning generic products, and the partial replacement upon bid-non-winning original products [10,44–46]. The change of manufacturer-level structure following the VBP policy reflects the reshaping of physicians' prescription preferences, suggesting that the VBP policy achieves cost-saving by severing the grey profit chain between hospitals and pharmaceutical enterprises and guiding the prescription behavior to return to the clinical needs of patients, which is a good attempt at value-based purchasing strategy [47].

## Price reduction serves secondary driver of drug expenditure reduction under the VBP policy

The most direct outcome of VBP policy implementation is the prominent price reduction of the bid-winning products, with an average price cut of approximately 50% per batch. In this study, the expenditure index decomposition revealed that

the price index of policy-covered drugs decreased from 1.01 ("4+7" pilot cities) and 0.97 ("4+7" expansion regions) to 0.69 and 0.62 following the VBP policy. It echoes the findings of previous studies that observed the general decline in the overall price level of VBP INNs [48–50]. The decline extent of the price index was secondary only to the manufacturer structure index, indicating that the absolute price reduction level played a secondary role in the decline of policy-covered drug expenditures.

It is noteworthy that a slight increment in the price index of alternative drugs was observed in the pilot cities following the implementation of the VBP policy, in line with the finding of Long et al. [50]. This might be related to insufficient monitoring and control of alternative drug use in hospitals during the early policy implementation stage, as we found that the price index increase was no longer present in the "4+7" expansion period. To achieve an integrated governance effect of policy, it is necessary to simultaneously consider relevant products in the same therapeutic field in VBP practice.

## Utilization quantity increase restrains VBP drug expenditure reduction and drives alternative drug expenditure reduction

This study found that, under the VBP policy, the quantity index of policy-covered drugs increased from 1.34 (pilot cities) and 1.96 (expansion regions) to 2.58 and 2.98, which contributed to the increase of corresponding drug expenditures. On the one hand, it is mainly related to the improvement of drug accessibility due to price cuts of bid-winning products, resulting in the release of patient demand and increased utilization of policy-covered drugs [44,51]. On the other hand, there is a certain degree of drug consumption increase induced by policy assessment [37]. It is suggested to be vigilant against the potential for hospital medication practices to swing to the opposite extreme under the VBP policy, from "economic (rebate) incentive" to "policy assessment incentive" [37,52]. Therefore, it is essential to optimize the VBP policy assessment method in hospitals and strengthen the intelligent monitoring of prescription big data to identify, control, and reduce unnecessary drug use and expenses.

We also observed a significant post-policy increase in the utilization of alternative drugs, driving up corresponding expenditures. More prominently, a study on early VBP policy attempts found that county public hospitals procured more antibiotics and a greater number of expensive antibiotics after policy implementation, leading to an increased total expenditure of antibiotics [53]. Alternative drugs are not directly affected by the VBP policy but are indirectly affected due to their "clinical substitutability" relationship with policy-covered drugs. The likely reason for the increase in alternative drug utilization might be that these policy-uncovered drugs still exist grey profit transmission space between pharmaceutical enterprises and hospitals [36], which needs to be solved by further expanding the scope of the VBP list and monitoring the use of alternative drugs.

## Limitations

Several potential limitations should be mentioned in this study. Firstly, as of April 2024, China's VBP practice at the national level has progressed to the ninth batch. However, limited by the observation duration of available data, this study only analyzed the implementation of the first VBP batch ("4+7" pilot and "4+7" expansion) using data from 2018 to 2020, which presents a limitation in terms of insufficient timeliness. Despite this, we believe that the trends and characteristics of drug expenditure changes following VBP policy should have consistent laws among different policy batches under big data [51]. This study included medical institution procurement data from multiple regions across the country, which might provide references for subsequent research and international policy practice. Secondly, this study is confined to data from public medical institutions. Under the trend of normalized VBP practice, retail drugstores and private hospitals are also gradually entering the scope of VBP, while relevant data are not within the observation of this study. In the future, it is necessary to carry out further analysis using full-channel data to enhance the completeness of research findings. Thirdly, potential limitations regarding the efficacy of causal inference should be considered. This study employed the time-varying DID method for causal inference, selected based on the policy characteristics of staggered implementation at different

time points. However, some scholars have noted limitations of the time-varying DID in estimating the average treatment effect of multi-period interventions [54,55], as its estimates can be regarded as weighted averages of multiple standard DID estimates [27]. This might lead to ambiguity in the interpretation of the time-varying DID coefficients. Although the robustness test of standard DID analyses has confirmed the reliability of the primary findings derived from time-varying DID, caution remains necessary when interpreting and comparing the estimated coefficients.

## Conclusion

China's VBP policy substantially reduced expenditures for both policy-covered drugs and their therapeutic categories, indirectly pointing to the relief of patients' financial burden. However, spillover effects increased alternative drug spending, partially weakening the policy effect on expenditure savings. The policy impact varied regionally and institutionally, showing greater efficacy in underdeveloped areas and secondary/tertiary hospitals. Drug expenditure changes drove by multiple factors: (1) the optimization of drug use structure due to manufacturer-level market displacement (primary driver), (2) the absolute price reduction caused by VBP policy (secondary driver), and (3) utilization increases that limited policy-covered drug savings while boosting alternative drug spending. Policy improvements should adopt holistic monitoring of policy-covered and uncovered drugs to control spillovers, while promoting high-quality generic substitution to enhance market efficiency.

## Supporting information

**S1 Table. General information of observation regions.** GDP, gross domestic product; CNY, Chinese yuan.
(PDF)

**S2 Table. General information of included drugs.**
(PDF)

**S3 Table. Drug expenditure changes by each pilot city.** VBP, volume-based procurement; INN, international nonproprietary name. GR1, the increment in 2019 against 2018; GR2, the increment in 2020 against 2019.
(PDF)

**S4 Table. Drug expenditure changes by each expansion province.** VBP, volume-based procurement; INN, international nonproprietary name. GR1, the increment in 2019 against 2018; GR2, the increment in 2020 against 2019.
(PDF)

**S5 Table. Parallel trend test for VBP drugs.** PHCs, primary healthcare centers.
(PDF)

**S6 Table. Parallel trend test for alternative drugs.** PHCs, primary healthcare centers.
(PDF)

**S7 Table. Parallel trend test for all observed drugs.** PHCs, primary healthcare centers.
(PDF)

**S8 Table. Robustness test by standard DID estimation.** VBP, volume-based procurement; INN, international nonproprietary name; *CI*, confidence interval; PHCs, primary healthcare centers; ATC, anatomical therapeutic and chemical.
(PDF)

**S9 Table. Robustness test by excluding potentially confounding time points.** VBP, volume-based procurement; INN, international nonproprietary name; *CI*, confidence interval; PHCs, primary healthcare centers; ATC, anatomical therapeutic and chemical.
(PDF)

**S1 Fig. Monthly trend of drug expenditure among (A) tertiary hospital, (B) secondary hospital, and (C) PHCs in pilot cities; (D) tertiary hospital, (E) secondary hospital, and (F) PHCs in expansion regions.**
(PDF)

## Author contributions

**Conceptualization:** Ying Yang, Lei Zhou, Zongfu Mao, Furong Wang.

**Formal analysis:** Ying Yang, Yuanhui Duan, Lei Zhou, Sisheng Gan.

**Funding acquisition:** Ying Yang, Furong Wang.

**Investigation:** Ying Yang, Yuanhui Duan, Lei Zhou, Sisheng Gan.

**Methodology:** Ying Yang, Yuanhui Duan.

**Supervision:** Zongfu Mao, Furong Wang.

**Writing – original draft:** Ying Yang, Yuanhui Duan, Lei Zhou.

**Writing – review & editing:** Ying Yang, Yuanhui Duan, Lei Zhou, Sisheng Gan, Zongfu Mao, Furong Wang.

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
