## [Decision Letter · Decision Letter 0]

1 Jul 2025

Dear Dr. Yang,

Thank you for submitting your manuscript to PLOS ONE. After careful consideration, we feel that it has merit but does not fully meet PLOS ONE’s publication criteria as it currently stands. Therefore, we invite you to submit a revised version of the manuscript that addresses the points raised during the review process.

We look forward to receiving your revised manuscript.

Kind regards,

Eric Nyarko, MPhil., Ph.D., MPH

Academic Editor

PLOS ONE

Journal Requirements:

3. In this instance it seems there may be acceptable restrictions in place that prevent the public sharing of your minimal data. However, in line with our goal of ensuring long-term data availability to all interested researchers, PLOS’ Data Policy states that authors cannot be the sole named individuals responsible for ensuring data access (http://journals.plos.org/plosone/s/data-availability#loc-acceptable-data-sharing-methods).

“This study was supported by the National Natural Science Foundation of China (Grant number: 72404098), the China Postdoctoral Science Foundation (grant number: 2024M761028), the Postdoctor Project of Hubei Province (grant number: 2024HBBHCXB019), the Postdoctoral Fellowship Program of CPSF (Grant number: GZC20240534), and the Undergraduate Innovation Program of Huazhong University of Science and Technology (Grant Number: X202410487082). The funding body played no part in the study design, collection, analysis or interpretation of data, the writing of the manuscript, or the decision to submit the manuscript for publication.”

Reviewers' comments:

Reviewer's Responses to Questions

**Comments to the Author**

1. Is the manuscript technically sound, and do the data support the conclusions?

Reviewer #1: Yes

Reviewer #2: Yes

2. Has the statistical analysis been performed appropriately and rigorously?

Reviewer #1: Yes

Reviewer #2: Yes

3. Have the authors made all data underlying the findings in their manuscript fully available?

Reviewer #1: Yes

Reviewer #2: Yes

4. Is the manuscript presented in an intelligible fashion and written in standard English?

Reviewer #1: Yes

Reviewer #2: Yes

Reviewer #1: The manuscript was technically sound, and the data support the conclusions.

The statistical analysis has been performed appropriately and rigorously.

The authors have provided all data found in their manuscript.

The manuscript is fairly complete and clear, and the English is acceptable.

Reviewer #2: The topic is timely and policy-relevant, and the analytical framework is broadly sound.

1. While the time-varying DID model is appropriate for estimating policy effects, the lack of a clean control group and possible concurrent external shocks (e.g., the early phase of the COVID-19 pandemic) should be more explicitly discussed as threats to causal inference.

2. The decomposition method and its interpretation are generally well explained, but additional clarification (perhaps a table or schematic) would help non-specialist readers.

3. Consider briefly elaborating on how the alternative drugs were selected (e.g., were they matched by ATC level or based on clinical judgment?).

4. Some figures and tables could benefit from clearer labeling (e.g., T0–T3 time periods should be defined in the figure captions themselves)

**Do you want your identity to be public for this peer review?** For information about this choice, including consent withdrawal, please see our Privacy Policy

Reviewer #1: No

Reviewer #2: No

---

## [Author Response · Author response to Decision Letter 1]

7 Jul 2025

Dear Eric Nyarko,

We appreciate your suggestions and reviewers’ kind comments on our submission entitled “Pharmaceutical expenditures changes under the volume-based procurement policy: effect and influencing factors” (PONE-D-25-17509). We have addressed all issues raised by the editor and reviewers and have revised the manuscript accordingly. The changes are marked with yellow highlighting in the resubmission version to facilitate the review process. Please find below the point-to-point responses.

All my best,

Ying Yang, PhD

Huazhong University of Science and Technology

Response to Journal Requirements

Response:

Many thanks to the editor for the reminder. We have checked the manuscript style when submitting revisions.

Response:

Thank you very much for the reminder. We confirm that the “Funding Information” section of this study is as follows (Lines 23-27 in Revised Manuscript with Track Changes):

Funding information

This study was supported by the National Natural Science Foundation of China (Grant number: 72404098), the China Postdoctoral Science Foundation (grant number: 2024M761028), the Postdoctor Project of Hubei Province (grant number: 2024HBBHCXB019), and the Postdoctoral Fellowship Program of CPSF (Grant number: GZC20240534).

3. In this instance it seems there may be acceptable restrictions in place that prevent the public sharing of your minimal data. However, in line with our goal of ensuring long-term data availability to all interested researchers, PLOS’ Data Policy states that authors cannot be the sole named individuals responsible for ensuring data access (http://journals.plos.org/plosone/s/data-availability#loc-acceptable-data-sharing-methods).

Response:

Thank you very much for the reminder. We confirm that the “Data Availability Statement” section of this study is as follows (Lines 15-20 in Revised Manuscript with Track Changes):

Data Availability Statement

The data that support the findings of this study are owned and managed by the National Health Commission (NHC) of the People’s Republic of China and are not publicly available due to institutional policies. The authors do not have the authority to share the data. Data access may be granted upon reasonable request through the official website (https://www.nhc.gov.cn/) or contact number (+8610-68792114) of NHC, subject to institutional approval.

“This study was supported by the National Natural Science Foundation of China (Grant number: 72404098), the China Postdoctoral Science Foundation (grant number: 2024M761028), the Postdoctor Project of Hubei Province (grant number: 2024HBBHCXB019), the Postdoctoral Fellowship Program of CPSF (Grant number: GZC20240534), and the Undergraduate Innovation Program of Huazhong University of Science and Technology (Grant Number: X202410487082). The funding body played no part in the study design, collection, analysis or interpretation of data, the writing of the manuscript, or the decision to submit the manuscript for publication.”

Response:

Many thanks to the editor for the reminder. We have included the “Funding information” in the resubmitted cover letter. As our response to comment 2 above, the “Funding information” section of this study was confirmed as follows (Lines 23-27 in Revised Manuscript with Track Changes):

Funding information

This study was supported by the National Natural Science Foundation of China (Grant number: 72404098), the China Postdoctoral Science Foundation (grant number: 2024M761028), the Postdoctor Project of Hubei Province (grant number: 2024HBBHCXB019), and the Postdoctoral Fellowship Program of CPSF (Grant number: GZC20240534).

Response:

Thank you very much for the reminder. We have added the “Supporting Information” section (Lines 433-454 in Revised Manuscript with Track Changes).

Response to Reviewers’ Comments

Reviewer #1: This paper summarized the changes in expenditure on policy-covered and uncovered drugs under China’s volume-based procurement and analyzed the driving factors and constraints influencing these changes. Overall, the research topic of this paper was scientific and practical, with a wide range of data and reliable analyses. The results and conclusions can provide useful references for improving the policy. The following suggestions are recommended to enhance the quality of this article:

1. Abstract lines 56-57: Relevant data should be provided.

Response:

Thank you very much for your reminder. We have made corresponding revisions in the Abstract section as follows (Line 44-46 in Revised Manuscript with Track Changes):

The decrease of VBP drug expenditures showed a trend of tertiary hospital (β=-0.64, p<0.001) > secondary hospital (β=-0.57, p<0.001) > primary healthcare centers (β=-0.39, p<0.001)

2. Both the abstract and the main text mention “large hospitals” several times. What is the specific definition of this term?

Response:

Thank you very much for pointing out the issue of unclear definition. To eliminate the ambiguity, we have directly used the explicit terms “secondary and tertiary hospitals” instead of “large hospitals” in the revised manuscript. The corresponding changes are marked in Lines 45, 330, and 426 in Revised Manuscript with Track Changes.

3. Abstract: The correspondence between “policy-covered drugs and uncovered drugs” and “VBP drugs and alternative drugs” should be noted.

Response:

Thanks very much for the helpful comments. We have made some revision in the Abstract section (Line 37 in Revised Manuscript with Track Changes), trying to make the meaning clear. As follows:

this study included 25 policy-covered VBP drugs and 99 policy-uncovered alternative drugs as study samples.

In addition, in the Materials and Methods section of the main text, we have added some content to clarify the relationship between “VBP drugs and alternative drugs” and “policy-covered drugs and uncovered drugs”. As follows (Lines 124, 129-130 in Revised Manuscript with Track Changes):

(1) VBP drugs. It represents the scope of drugs covered by the VBP policy.

(2) Alternative drugs. It represents the scope of drugs related to VBP policy but not directly covered by the policy.

4. Abstract line 65: The conclusion states that “the VBP policy reduced drug expenditures and is beneficial in relieving patient medication burden.” However, the study does not seem to have a clear corresponding indicator. The discussion section mentioned that other micro-level studies have reached similar conclusions, but this was not a direct conclusion of this study. The conclusion should be expressed more cautiously.

Response:

Thank you very much for the suggestion. To ensure the rigor of study conclusion, we have taken your suggestion to drop the phrase “relieving patient medication burden” in the revised manuscript. The corresponding changes are marked in Lines 52-54 in Revised Manuscript with Track Changes, as follows:

VBP policy effectively promoted the decline of total drug expenditures, primarily through manufacturer-level market displacement and the absolute price reduction.

5. Introduction: The innovation of this study is emphasized as analyzing from the perspective of hospital drug use. Please explain in detail how this perspective differs from existing macro- and micro-level studies, and explain how analysis from this perspective can compensate for the shortcomings of existing studies, thereby more effectively supporting the objectives and significance of this study.

Response:

Many thanks to the reviewer for pointing out our oversight. We fully agree with your perspective that explicitly stating the limitations of previous studies and highlighting the distinctive aspects of the present study in the Introduction section is indeed crucial. We have added relevant content accordingly in the last paragraph of the Introduction section. As follows (Lines 96-100 in Revised Manuscript with Track Changes):

In summary, the large number of existing research on drug expenditures under VBP policy implementation, on the one hand, lack reliable evidence when involving the scope of drugs not covered by the policy; on the other hand, no studies have systematically investigated the elements influencing drug expenditure changes from the perspective of hospital drug use. This study tries to make up for the above two aspects.

6. Methods: The introduction mentioned that this study was analyzed from the perspective of hospital drug use, and the discussion referred to drug expenditure as macro-level expenditure. However, the specific definition and statistical scope of drug expenditure were not clearly stated in the methods section. This should be explicitly explained in the methods section to avoid ambiguity.

Response:

We sincerely appreciate the reviewer’s valuable suggestions. In response to this comment, we have added a new paragraph in the “Difference-in-difference model” section to clarify the definition of drug expenditure indicator as well as its statistical method. The corresponding modifications are highlighted in Lines 158-167 in Revised Manuscript with Track Changes:

The drug expenditure indicator specifically refers to medical institutions’ procurement expenditures of a defined range of drugs, which is distinct from the medication costs of individual patients occurred during outpatient or inpatient visits. However, under the zero-markup policy that mandates identical procurement and retail prices for drugs in public medical institutions [25], and given procurement regulations requiring all drugs to be procured through provincial drug centralized bidding and procurement platform [26], drug procurement expenditures can essentially be considered as the aggregate sum of medication costs across all patient visits. Following the fundamental principle of expenditure equals price multiplied by quantity, the expenditure (Eit) of drug i during the observation period t was calculated as the product of the price (Pit) and quantity (Qit). The formula is expressed as follows:

"E" _"it" "=" ∑_"t=1" ^"n" ▒∑_"i=1" ^"n" ▒〖"P" _"it" "×" "Q" _"it" 〗 (1)

7. Data sources line 125: Please provide a detailed definition of the “36 time points”?

Response:

Thanks very much for the reminder. For clarity, we have revised the relevant wording as follows (Lines 115-117 in Revised Manuscript with Track Changes):

This study extracted data spanning a total of 36 months from January 2018 to December 2020, with the month serving as the smallest observational time unit, encompassing 36 observation time points.

8. Samples: The number of alternative drugs, drug generic names, drug products, and involved drug manufacturers should also be specified separately.

Response:

We thank the reviewer for pointing out this issue. We have added relevant information in the revised manuscript. As follows (Lines 139-141 in Revised Manuscript with Track Changes):

A total of 99 alternative INNs were included in the analysis, covering 256 drug generic names, 3523 drug products, and involving 676 drug manufacturers.

9. Index decomposition method on drug expenditure: T0 and T1 were both stages where the policy has not been implemented. Please explain the basis for this division.

Response:

We appreciate the reviewer’s reminder. The reason for setting multiple observation time points before the policy implementation was to capture the pre-policy trend and attempt to control for confounding factors beyond the VBP policy. We have added relevant content in the revised manuscript. As follows (Lines 222-227 in Revised Manuscript with Track Changes):

Among them, T0 and T1 represent the pre-intervention periods of “4+7” pilot, while T2 and T3 denote the post-intervention periods; T0, T1, and T2 represent the pre-intervention periods of “4+7” expansion, with T3 marking the post-intervention period. We designed multiple observation periods prior to policy implementation to monitor pre-intervention trends and mitigate potential confounding factors beyond the VBP policy.

10. The analysis in the article involved many classifications, such as region, time, VBP drugs/alternative drugs, original drugs/generic drugs, medical institution grading, and drug ATC classification. It is recommended to describe the analysis framework, classification basis, and definitions of each category in the methods section, especially the analysis approach for subgroup analysis, to enhance the readability of the results section.

Response:

Thanks very much for the valuable comment, and we fully agree with your suggestion.

In response to this, first, we have added a detailed description of the overall analytical framework of this study, which includes considerations regarding the classification of time periods, geographical regions, and the scope of drugs. As follows (Lines 149-155 in Revised Manuscript with Track Changes):

This study adopts a comparative analysis approach overall. Firstly, in the time dimension, we divided time period before and after policy intervention to quantify changes and impacts caused by VBP policy implementation. Secondly, in the region dimension, we distinguished between “4+7” pilot cities and expansion regions, which represent two policy implementation phases with different implementation timelines, necessitating separate assessments and comparisons. Thirdly, in the drug scope dimension, we i

---

## [Decision Letter · Decision Letter 1]

30 Jul 2025

Pharmaceutical expenditures changes under the volume-based procurement policy: effect and influencing factors

PONE-D-25-17509R1

Dear Dr. Yang,

We’re pleased to inform you that your manuscript has been judged scientifically suitable for publication and will be formally accepted for publication once it meets all outstanding technical requirements.

Kind regards,

Eric Nyarko, BSc, MPhil, PhD, MPH

Academic Editor

PLOS ONE

Additional Editor Comments (optional):

Reviewers' comments:

Reviewer's Responses to Questions

**Comments to the Author**

Reviewer #1: All comments have been addressed

Reviewer #2: All comments have been addressed

2. Is the manuscript technically sound, and do the data support the conclusions?

Reviewer #1: Yes

Reviewer #2: Yes

3. Has the statistical analysis been performed appropriately and rigorously?

Reviewer #1: Yes

Reviewer #2: Yes

4. Have the authors made all data underlying the findings in their manuscript fully available?

Reviewer #1: Yes

Reviewer #2: Yes

5. Is the manuscript presented in an intelligible fashion and written in standard English?

Reviewer #1: Yes

Reviewer #2: Yes

Reviewer #1: (No Response)

Reviewer #2: Thank you for your sincere response and your revision improved the manuscript.

**Do you want your identity to be public for this peer review?** For information about this choice, including consent withdrawal, please see our Privacy Policy

Reviewer #1: No

Reviewer #2: No

---

## [Editor Report · Acceptance letter]

PONE-D-25-17509R1

PLOS ONE

Dear Dr. Yang,

I'm pleased to inform you that your manuscript has been deemed suitable for publication in PLOS ONE. Congratulations! Your manuscript is now being handed over to our production team.

Kind regards,

on behalf of

Dr. Eric Nyarko

Academic Editor

PLOS ONE